# Defining a recovery-oriented cascade of care for opioid use disorder: A community-driven, statewide cross-sectional assessment

Jesse L. Yedinak[1], William C. Goedel[1], Kimberly Paull[2], Rebecca Lebeau[2], Maxwell S. Krieger[1], Cheyenne Thompson[2], Ashley L. Buchanan[3], Tom Coderre[4], Rebecca Boss[5], Josiah D. Rich[1,6], Brandon D. L. Marshall[1]*

1 Department of Epidemiology, School of Public Health, Brown University, Providence, Rhode Island, United States of America, 2 Executive Office of Health and Human Services, State of Rhode Island, Cranston, Rhode Island, United States of America, 3 Department of Pharmacy Practice, College of Pharmacy, University of Rhode Island, Kingston, Rhode Island, United States of America, 4 Office of the Governor, State of Rhode Island, Providence, Rhode Island, United States of America, 5 Department of Behavioral Healthcare, Developmental Disabilities and Hospitals, State of Rhode Island, Cranston, Rhode Island, United States of America, 6 Department of Medicine, Warren Alpert Medical School, Brown University, Providence, Rhode Island, United States of America

* brandon_marshall@brown.edu

**Data Availability Statement:** Aggregated data or publicly available survey results were used whenever possible. Data for Stage 0 are available

## Abstract

### Background

In light of the accelerating and rapidly evolving overdose crisis in the United States (US), new strategies are needed to address the epidemic and to efficiently engage and retain individuals in care for opioid use disorder (OUD). Moreover, there is an increasing need for novel approaches to using health data to identify gaps in the cascade of care for persons with OUD.

### Methods and findings

Between June 2018 and May 2019, we engaged a diverse stakeholder group (including directors of statewide health and social service agencies) to develop a statewide, patient-centered cascade of care for OUD for Rhode Island, a small state in New England, a region highly impacted by the opioid crisis. Through an iterative process, we modified the cascade of care defined by Williams et al. for use in Rhode Island using key national survey data and statewide health claims datasets to create a cross-sectional summary of 5 stages in the cascade. Approximately 47,000 Rhode Islanders (5.2%) were estimated to be at risk for OUD (stage 0) in 2016. At the same time, 26,000 Rhode Islanders had a medical claim related to an OUD diagnosis, accounting for 55% of the population at risk (stage 1); 27% of the stage 0 population, 12,700 people, showed evidence of initiation of medication for OUD (MOUD, stage 2), and 18%, or 8,300 people, had evidence of retention on MOUD (stage 3). Imputation from a national survey estimated that 4,200 Rhode Islanders were in recovery from OUD as of 2016, representing 9% of the total population at risk. Limitations included use of self-report data to arrive at estimates of the number of individuals at risk for OUD and using

through the National Survey on Drug Use and Health on the website https://nsduhweb.rti.org. For Stage 1 we partnered with third party state institutions who provided us with aggregated data, we therefore do not have permission to share anything other than the aggregated estimates that were in the manuscript. Aggregated data for Stages 2 and 3 are available on the webpage https://preventoverdoseri.org/medication-assisted-therapy/. For Stage 4 we utilized results from the National Recovery Survey conducted in 2016 by Kelly, Bergman, Hoeppner, Vilsaint, and White (2017) https://www.recoveryanswers.org/media/national-addiction-recovery-study/.

**Funding:** JDR and BDLM are supported by the COBRE on Opioids and Overdose funded by the National Institute of General Medical Sciences of the National Institutes of Health under grant number P20GM125507 https://www.nigms.nih.gov/. Funding for Rhode Island's statewide overdose surveillance website, www.PreventOverdoseRI.org, is provided to the Rhode Island Department of Health from the Centers for Disease Control & Prevention under grant number CDC RFA-CE15-1501 https://www.cdc.gov/drugoverdose/od2a/index.html. The funders had no role in study design, data collection and analysis, decision to publish, or preparation of the manuscript.

**Competing interests:** The authors have declared that no competing interests exist.

**Abbreviations:** APCD, all-payer claims database; BHOLD, Behavioral Health On-Line Database; MOUD, medication for opioid use disorder; NSDUH, National Survey on Drug Use and Health; OUD, opioid use disorder; PDMP, Prescription Drug Monitoring Program; R-DAS, Restricted-use Data Analysis System.

a national estimate to identify the number of individuals in recovery due to a lack of available state data sources.

## Conclusions

Our findings indicate that cross-sectional summaries of the cascade of care for OUD can be used as a health policy tool to identify gaps in care, inform data-driven policy decisions, set benchmarks for quality, and improve health outcomes for persons with OUD. There exists a significant opportunity to increase engagement prior to the initiation of OUD treatment (i.e., identification of OUD symptoms via routine screening or acute presentation) and improve retention and remission from OUD symptoms through improved community-supported processes of recovery. To do this more precisely, states should work to systematically collect data to populate their own cascade of care as a health policy tool to enhance system-level interventions and maximize engagement in care.

## Author summary

### Why was this study done?

- In the US, drug overdose represents a leading cause of accidental death. In light of this growing epidemic, frameworks are needed to understand how to improve health systems to identify and engage individuals with substance use disorders in evidence-based treatment modalities.

- Cascades of care have been used to track and improve population health outcomes for multiple complex health conditions by encouraging data-driven policy decisions to adapt and strengthen systems of care for how these conditions are managed, but few cascades of care are available for use for local jurisdictions addressing opioid use disorder (OUD).

### What did the researchers do and find?

- We engaged a group of stakeholders—local experts on opioid use and its consequences, leaders from state agencies governing health and social services, directors of nongovernmental organizations providing health and social services to people living with OUD, and community advocates with lived experiences of OUD and recovery—to adapt and define a cascade of care for OUD for use in Rhode Island.

- The stakeholder engagement process resulted in a cascade of care with 5 stages, beginning with individuals at risk for OUD (stage 0), continuing to individuals who are diagnosed with OUD (stage 1) and establish engagement with a medication-based treatment plan (stage 2), and ending with continuous engagement with this treatment plan (stage 3) and recovery (stage 4).

- Using national survey estimates and statewide administrative claims databases, we found that 26,000 Rhode Islanders were diagnosed with OUD (stage 1) in 2016, 12,700 people showed evidence of treatment initiation (stage 2), and 8,300 had evidence of continuous engagement with treatment for at least 6 months (stage 3). Based on a national

survey estimate, about 4,200 individuals are estimated to have achieved recovery from OUD using medications (stage 4).

## What do these findings mean?

- Engagement with a diverse group of stakeholders can result in the development of a cascade of care to assess and measure the success of statewide health systems in delivering interventions to address opioid-related harms. The cascade of care can be used as a framework to strengthen health systems that may result in reductions in the number of individuals at risk for OUD and increases in the number of individuals with OUD who are able to achieve long-term recovery.

- The estimates of the numbers of individuals in each stage represent a static "snapshot" and are considered preliminary; further efforts are needed to fine-tune these proportions. For example, limitations included having to use the definition of recovery and estimates used in the National Recovery Survey, as there are currently no statewide data sources for measuring recovery. Further research is needed to understand how to best define and operationalize this stage at a statewide level.

## Introduction

As the drug overdose crisis in the US continues to accelerate [1], new strategies are needed to address the epidemic and to more efficiently engage and retain individuals with substance use disorder in care. Applying a cascade of care framework offers a novel approach to curb this unrelenting crisis of drug-related harms by identifying novel points for intervention at the system level [2]. Cascades of care have been used to track and improve population health outcomes for multiple complex health conditions [3–14], with the most well-known and visible of such being the continuum of care for HIV infection [4,9,10,12–14]. Applying this continuum of care framework has drastically improved health outcomes among people living with HIV infection, encouraged data-driven policy decisions, and brought about revolutionary system-level changes worldwide in how HIV infection is managed across the course of the disease [12–14].

Researchers and policymakers have called for a cascade of care framework to be applied to understand gaps in treatment engagement among those experiencing substance use disorder, and opioid use disorder (OUD) in particular [2,15,16]. Specifically, a new framework for assessing the availability and quality of care for OUD has recently been defined by Williams and colleagues using metrics defined by the National Quality Forum and the Agency for Healthcare Research and Quality [17]. Preliminary evidence suggests that improvements in health delivery processes at a system level are linked to improved health outcomes and lower mortality among patients with OUD [18]. Defining a cascade of care is a critical first step in helping local jurisdictions establish and utilize their data resources to inform policy and promote interventions that protect against the potentially deadly harms of OUD when left untreated [15,19]. Herein, we aim to adapt and apply the cascade of care defined by Williams and colleagues [20] for the use of medication for OUD (MOUD) [21], using key national and statewide datasets for Rhode Island, a small state in New England, a region of the US heavily affected by the opioid crisis [1].

## Methods

### Study setting

Rhode Island is an ideal setting for local adaptation of a cascade of care framework for OUD, given ongoing cross-agency, statewide policy efforts to curb the overdose crisis, and the availability of statewide health claims data to evaluate OUD care [22]. Since 2014, state agencies and community leaders have engaged in data sharing and multidimensional surveillance of the opioid crisis [23]. Beginning in 2015, the statewide Overdose Prevention and Intervention Task Force has led the creation and implementation of the Overdose Prevention and Intervention Action Plan [22], designed to monitor key health metrics and guide data-driven program delivery to help end the overdose crisis in the state [22].

### Stakeholder engagement

Given that a large amount of programmatic and surveillance data were already being collected and shared on the state's publicly accessible overdose data dashboard, PreventOverdose, RI (https://preventoverdoseri.org/), to assess progress towards the goals of the action plan [23], key state agency partners engaged with a team at Brown University beginning in June 2018 to begin developing a statewide cascade of care for people living with OUD. These evaluation activities did not meet the federal definition of research, and, as such, ethics approval was not required. The stakeholder group involved 28 members, including local experts on opioid use and its consequences, leaders from state agencies governing health and social services, directors of nongovernmental organizations providing health and social services to people living with OUD, and community advocates with lived experiences of OUD and recovery. This group met 7 times between June 2018 and May 2019 and developed the proposed cascade of care for Rhode Island through an iterative process. Prior to discussions designating the specific stages of the cascade of care, the stakeholder group agreed on a set of shared terminology (Table 1) and identified shared values to guide the framework development process. There were no prespecified plans for defining the stages of the cascade of care or the data sources

**Table 1. Glossary of terms guiding the framework development process.**

| Term | Definition |
|---|---|
| Cascade of care | A cascade of care [2,14,15,24,25] is a conceptual framework to guide and track patients (people) over time through stages of medical care for a particular disease or condition, allowing for identification of key points or places to intervene and improve health outcomes [2]. |
| Opioid use disorder (OUD) | We use OUD as the overarching medical condition to define the population of people measured in the cascade of care through an OUD diagnosis. OUD is also used to identify the healthcare systems (places) and transition points (processes) relevant for engaging people at each stage of the cascade [26]. |
| Criteria for OUD | We use Diagnostic and Statistical Manual of Mental Disorders (DSM-5) criteria for diagnosing OUD, which is defined by loss of control of opioid use, risky opioid use, impaired social functioning, tolerance, and withdrawal symptoms from opioids [27,28]. |
| Indicators | Indicators are the "units of service" or "process measures" in the datasets, which are the details that help us define the size of each stage in the cascade of care, as compared to other stages. They also allow us to assess the quality of referrals and retention from one stage to the next [18,29–32]. |
| Screening and assessment for OUD | Evidence-based screening instruments (secondary prevention) that can be used at point of care (such as a clinic) to help identify someone experiencing opioid misuse or OUD. Examples include the Screening, Brief Intervention, and Referral to Treatment (SBIRT) or the National Institute on Drug Abuse (NIDA)–modified Alcohol, Smoking and Substance Involvement Screening Test (ASSIST) [33,34]. |

used to estimate the current state of the cascade of care prior to this stakeholder engagement process. All methods discussed below have been described in line with the Strengthening the Reporting of Observational Studies in Epidemiology (STROBE) Statement (see S1 STROBE Checklist).

Later discussions designating the specific stages of the cascade of care were grounded in the framework proposed by Williams and colleagues in 2017, which was updated in 2019 [15,20]. This cascade includes 4 stages, beginning with stage 1, where individuals are already identified as having OUD or experiencing a nonfatal overdose [15,20]. The first stage, engagement in care, is defined as the proportion of individuals with OUD who receive specialty services in a given year [20]. The second stage, MOUD initiation, is defined as the percentage of individuals in care who receive MOUD at least once [20]. The third stage, retention, is defined as the proportion of individuals who receive MOUD who continue to do so for at least 180 days [20]. The fourth stage, remission, is defined as the proportion of retained individuals who no longer meet diagnostic criteria for OUD [20].

With this framework informing the foundation of the adapted cascade of care, stakeholders were presented with the available measures of healthcare access and quality to represent each stage [16,17] and discussed the feasibility of using these metrics to develop a "snapshot" of the current state of the cascade of care for OUD in Rhode Island in the year 2016. This process led to additional refinements to the cascade and finalization of the operational definitions of each stage while also helping to define the target population (e.g., those experiencing OUD) and the healthcare systems and data-sharing partners for the cascade [20,26]. By using national quality metrics, the cascade could be designed to enhance existing statewide policy and surveillance tools by (1) creating an annual measure of statewide engagement in care using available datasets, (2) setting targets and standards for improving linkages across stages of care, and (3) defining successful endpoints for treatment of OUD through consensus among the stakeholders [17,18,29,30].

## Results

### Identifying the guiding principles of a cascade of care for OUD

The stakeholder engagement process resulted in a set of guiding principles for the cascade of care. Through this community-driven process, stakeholders determined that a cascade of care for OUD should be

- **Measurable and achievable**: The cascade of care should track statewide progress in connecting people to high-quality OUD care by identifying meaningful stages of engagement in care and defining measurable targets for engagement in care at each of its stages.

- **Timely and dimensional** [35]: The cascade of care should measure statewide progress year over year using input from multiple datasets and systems of care. This includes measuring the estimated number of patients in each stage at regular intervals and assessing their progress in moving from one stage to the next.

- **Incremental**: The cascade of care should be used to increase impact at the population level. Through this lens, interventions moving larger groups of individuals to a subsequent stage should be prioritized, rather than moving small numbers of individuals immediately to the end stages of the cascade.

- **Inclusive**: The definition of successful progress through the cascade of care should be client-centered and inclusive of available evidence-based treatment modalities, patient requests, and provider opinions.

- **Voluntary**: Individuals counted in the cascade should be those who voluntarily entered treatment.

- **Equitable**: Comprehensive analyses of health delivery processes have the capacity to uncover patterns of previously unseen inequities across health systems, across demographic groups, among special populations, or across stages of care. Inequalities that appear as systemic barriers to successful progress along the cascade should be investigated and responded to as they arise.

## Defining the stages of the cascade of care for OUD

In the preliminary framework proposed by Williams and colleagues [20], "remission" is defined as a finite endpoint, similar in spirit to viral load suppression in the context of the continuum of care for HIV infection [9]. However, OUD was conceptualized by our stakeholder group as a chronic brain disease [36] (rather than as a chronic infection) that has the potential for both relapse and recovery, and where persons are subject to movement in and out of systems of care. As such, our stakeholder group sought to capture more than a single clinical endpoint for the final stage, given that clinical records indicating symptom remission, or absence of evidence of recurrence (e.g., an emergency department visit for a drug overdose) did not fully capture the spectrum of engagement in care for and recovery from OUD. Further, while retention on MOUD represents one element of successful treatment, stakeholders felt that measuring engagement in recovery support services [37,38], including those offered in outpatient healthcare settings and those offered by peers in the community, was critical for measuring long-term absence of OUD symptoms.

With this perspective, the long-term goal of the cascade of care developed by the stakeholder group (hereafter referred to as the Rhode Island Cascade of Care for Opioid Use Disorder) was 2-fold. Similar to the framework proposed by Williams and colleagues [20], the organization of the cascade reflects a primary prevention goal of reducing the number of individuals at risk for OUD (stage 0), while also increasing the number who remain active in their recovery through engagement with professional and peer-based support services (stage 4). The stages of the Rhode Island Cascade of Care for Opioid Use Disorder are defined below (Fig 1).

Stage 0 includes those who are considered at risk for OUD and represents the population who may benefit from targeted prevention efforts and/or early intervention services. Individuals in this stage may meet the diagnostic criteria for a clinical diagnosis of OUD at some point during their use, but many may not. Furthermore, this heterogeneous group may include treatment-naïve individuals as well as individuals who were at some point in recovery from OUD. Therefore, the long-term goal is to reduce the total population at risk for OUD over time through a combination of primary prevention efforts (e.g., improved opioid prescribing guidelines to prevent misuse of prescription opioids) as well as secondary prevention efforts (e.g., early intervention through routine screening and rapid referral) [20]. This stage was operationalized as the number of people who reported using heroin and/or misusing a prescription opioid (i.e., using a prescription opioid without a prescription or in any way other than as described by a doctor) in the past 12 months, consistent with measures in the National Survey on Drug Use and Health (NSDUH) [39].

Stage 1 represents those diagnosed with OUD. This stage is operationalized as the number of individuals with a medical claim tied to a diagnosis of OUD in a given year, based on existing national quality metrics [17].

Stage 2 represents unique individuals who have initiated MOUD. The national quality metrics suggest including only those individuals who have been on MOUD for more than 7 days

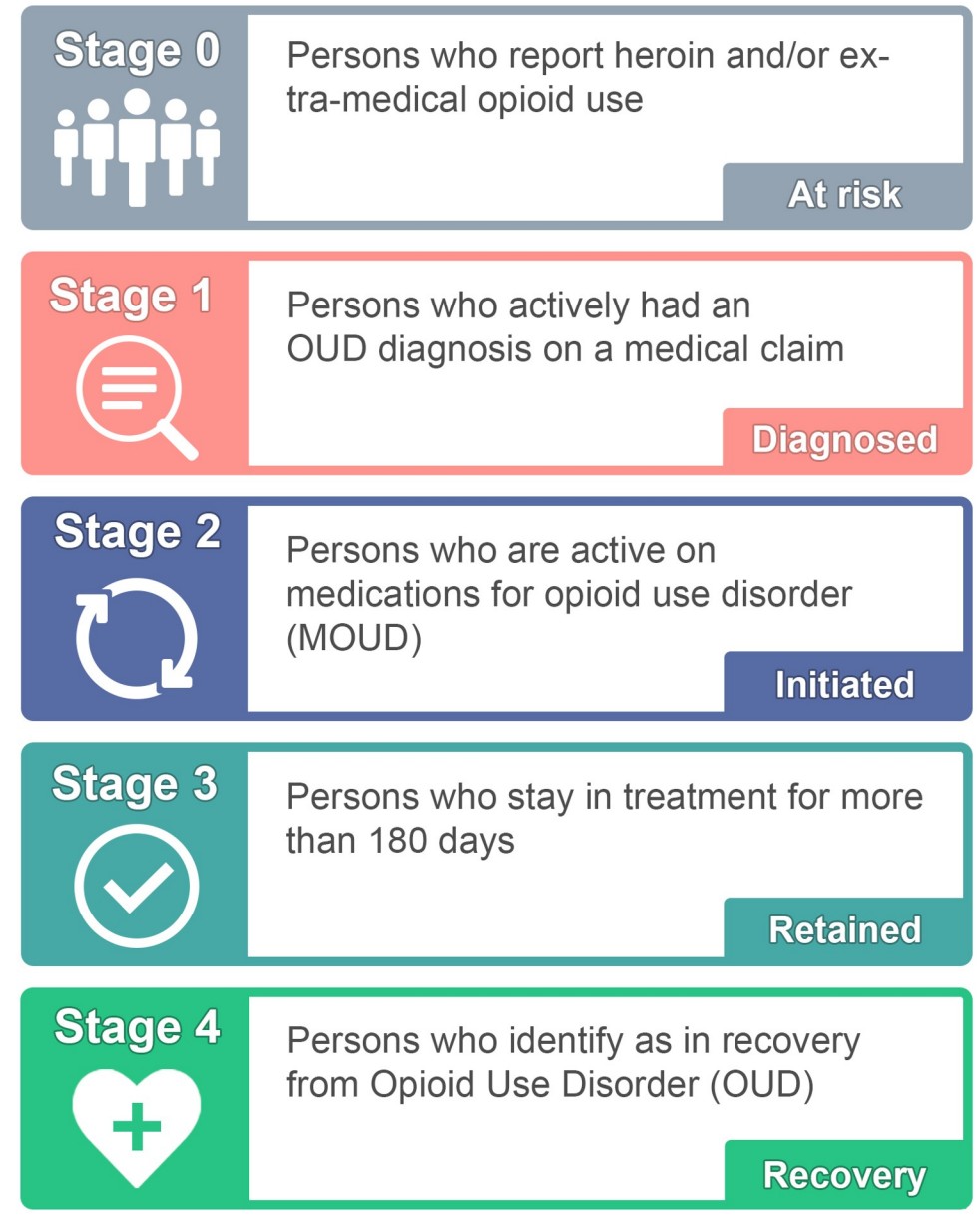

**Fig 1. Overview of the Rhode Island model for the cascade of care for opioid use disorder (OUD).** Credit: Maxwell Krieger, Brown University.

at one time (representing stabilization) but have been engaged for less than 180 days [29,30]; those who have been engaged for 7 days or less would remain in stage 1. Although we acknowledge other modalities of treatment for OUD, the stakeholders chose to focus on MOUD given strong evidence supporting its effectiveness [40–42] and the available datasets on prescribing and dispensing of these medications to persons with OUD.

Stage 3 represents those who are retained on MOUD. Based on existing national quality metrics [30,43], persons who are retained have engaged with their treatment plan for 180 days or longer at one time, without a gap of more than 7 days. The goal in moving

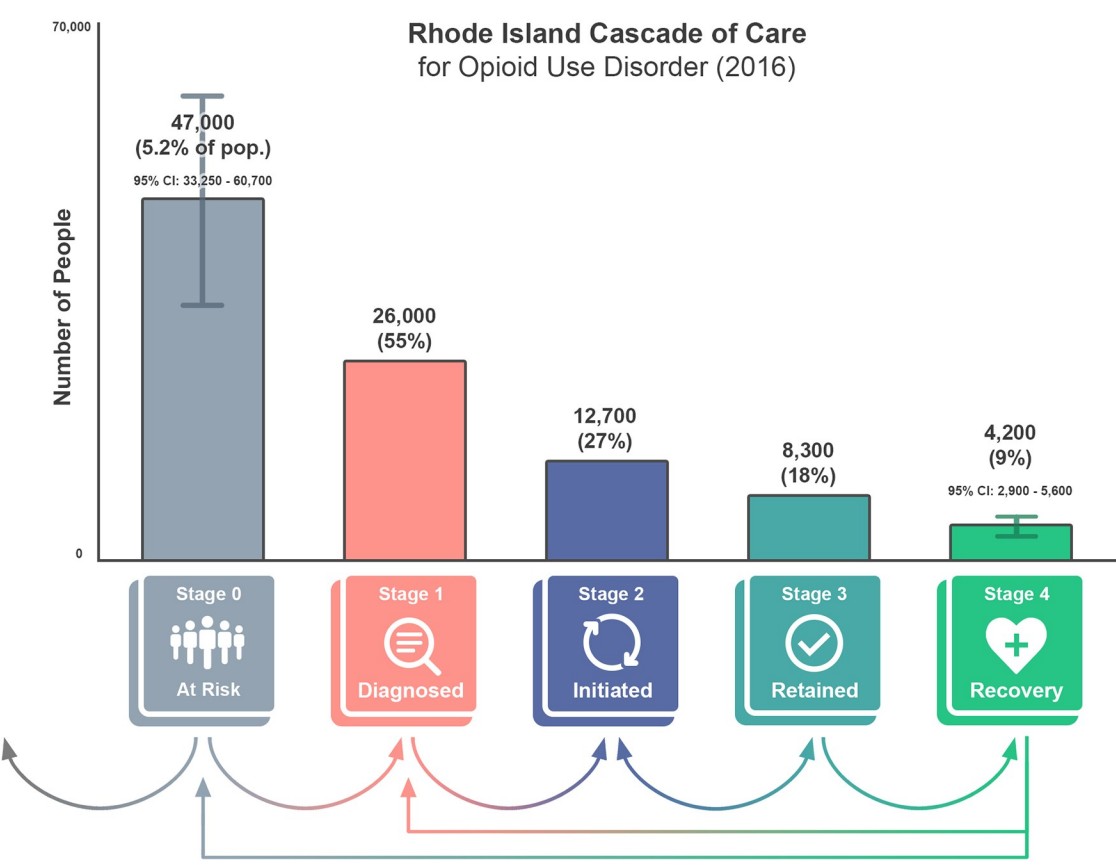

**Fig 2. Results for the Rhode Island Cascade of Care.** Stages 0 and 4 represent estimates from national survey data sources. Stage 1 represents statewide claims data from the HealthFacts RI all-payer claims database (APCD). Stages 2 and 3 represent combined estimates from the Rhode Island Prescription Drug Monitoring Program (PDMP) and the Behavioral Health On-Line Database (BHOLD), which include treatment claims for methadone and records for buprenorphine prescriptions. All estimates are approximate and considered preliminary. Credit: Maxwell Krieger, Brown University.

individuals to this stage is to support continuous engagement with treatment services for at least 6 months.

Stage 4 represents recovery from OUD. In the context of this cascade of care, recovery is defined as having achieved sustained remission from or resolution of symptoms of OUD using a MOUD-assisted pathway [37,38,44]. The goals for this stage are prevention of OUD symptom recurrence and support of active recovery through structural interventions (e.g., housing, employment programs) and engagement with recovery community centers and community-based peer support services. Stakeholders noted the challenges of measuring the size of this population in existing datasets, but noted that this stage may be measured by considering individuals who have engaged with MOUD for at least 180 days as synonymous with those who are in recovery in the absence of additional data, or by supporting primary data collection activities where self-reported recovery from OUD is captured in ongoing statewide health assessments.

The arrows at the bottom of Fig 2 represent possible transitions through the cascade in 1 year. For example, individuals who are diagnosed with OUD in a given year may initiate MOUD, be retained in care, and achieve recovery. Persons who meet the definition of recovery may become at risk again, thus transitioning back to stage 0 (if never formally diagnosed) or

stage 1 (if ever formally diagnosed). Some people may exit the population entirely without progressing beyond stage 0 (i.e., through cessation of drug use), as shown by the leftmost arrow.

## Preliminary estimation of the Rhode Island Cascade of Care for Opioid Use Disorder

A preliminary estimation of the number of individuals included in each stage of the Rhode Island Cascade of Care for Opioid Use Disorder is displayed in Fig 2.

Identifying the number of individuals included in stage 0 leverages data collected as part of NSDUH. This survey has been conducted by the US Department of Health and Human Services since 1971. Each year, about 70,000 people aged 12 years and older are interviewed to provide self-reported information on alcohol, tobacco, and drug use; mental health; and other health-related issues [39]. The Restricted-use Data Analysis System (R-DAS) was launched by the Substance Abuse and Mental Health Services Administration (SAMHSA) as an online analytic system that allows analysts to produce cross-tabulations using restricted-use NSDUH datafiles [39]. R-DAS allows for the creation of state-level estimates of select variables using revised weights and combining multiple years of data collection [44]. Using R-DAS, we calculated an estimate for the combined total number of people who reported heroin use and/or misuse of a prescription opioid in the past 12 months in Rhode Island. This population estimate combines data from 2015 and 2016 and, as such, is meant to be representative of the average annual population across the 2 years. Based on this analysis, we estimate that there were 47,000 (95% CI 33,250–60,700) people in Rhode Island in stage 0 in 2016. This estimate serves as the cross-sectional denominator for the Rhode Island Cascade of Care for Opioid Use Disorder.

Identifying the number of individuals included in stage 1 utilized data from HealthFacts RI, the state's APCD. The APCD stores information on enrollment, medical claims, pharmacy claims, and healthcare providers from privately insured individuals as well as Medicare and Medicaid recipients [45]. Following the Supreme Court ruling in Gobeille v. Liberty Mutual Insurance Company in 2016, the APCD does not include information on individuals with self-funded employee health plans (i.e., those who are self-insured), representing about 20% of Rhode Islanders. Individuals included in stage 1 included those with active claims in 2016 using ICD-9-CM codes 304.00–304.03 (opioid dependence), 304.70–304.73 (dependence combinations of opioid type drug with any other), and 305.50–305.53 (opioid abuse) and using ICD-10 codes F11.1x (opioid abuse), F11.2x (opioid dependence), and F11.9x (opioid use, unspecified). Based on this data source, we estimated that there were 26,000 people in Rhode Island in stage 1 in 2016, representing an estimated 55% of people at risk for OUD (stage 0).

Identifying the number of individuals in stage 2 utilized data from PDMP and BHOLD, 2 databases maintained by the Rhode Island Department of Health and the Rhode Island Department of Behavioral Healthcare, Developmental Disabilities and Hospitals, respectively [46,47]. Using these data, we calculated the number of unique individuals receiving methadone or buprenorphine for more than 7 days at one time in 2016. Based on these sources, we estimated that there were 12,700 people in Rhode Island in stage 2, representing an estimated 27% of people at risk for OUD (stage 0) and 49% of people diagnosed with OUD (stage 1).

Identifying the number of individuals in stage 3 also utilized data from PDMP and BHOLD [46,47]. Using these datasets, we identified unique individuals who were engaged with MOUD through a certified opioid treatment program or prescribed buprenorphine for at least 180 days without a gap of more than 7 days based on the date of first service. Individuals meeting the criteria for retention at any point in 2016 were included. Based on these data sources, we estimated that there were approximately 8,300 people in Rhode Island who progressed to stage

3 through the use of methadone or buprenorphine, representing an estimated 18% of people at risk for OUD (stage 0) and 65% of people who initiated MOUD (stage 2).

Identifying the number of individuals in stage 4 utilized data from the National Recovery Survey conducted in 2016 by Kelly et al. [37]. The National Recovery Survey was a national probability-based sample of adults in the US who self-identified as having resolved a significant problem with alcohol and other drugs [37]. Individuals who answered "Yes" to the question "Did you used to have a problem with drugs or alcohol, but no longer do?" were identified as being in recovery. Based on the survey response to this screening question, these individuals were estimated to represent 9.1% of the adult population of the US (95% CI 8.6%–9.6%). Among individuals who achieved recovery, 5.3% were estimated to have achieved recovery from an opioid problem (95% CI 3.8%–6.8%) using MOUD [38]. In applying these estimates to the adult population of Rhode Island, we estimated that there were 4,200 people (95% CI 2,900–5,600) in Rhode Island who achieved recovery through use of MOUD, representing 9% of people at risk for OUD and 50% of people retained on MOUD (stage 3).

## Discussion

Using a combination of national and statewide databases, we were able to generate a statewide, cross-sectional cascade of care for the treatment of OUD in Rhode Island beginning with those at risk for OUD and ending with those in recovery from OUD. Initial estimates of the number of individuals in each stage indicated that Rhode Island has high initiation and retention rates for engagement with MOUD following a diagnosis with OUD. This is consistent with 2017 data from the National Survey of Substance Abuse Treatment Services, which showed that Rhode Island had a rate of 419 per 100,000 people aged 18 years and older engaged with MOUD, ranking among the top 5 states in the nation [48]. However, the results also indicate that additional efforts, such as enhanced screening, are needed to identify those at risk and engage them in care to achieve the dual purpose of mitigating potential harms of opioid use and increasing opportunities for diagnosis and treatment of OUD symptoms. In addition, promoting and supporting continuous engagement with MOUD modalities may provide additional protection against the potential harms of opioid use among those who initiate MOUD but are not retained [49,50].

In the process of developing and defining the Rhode Island Cascade of Care for Opioid Use Disorder, stakeholders expressed the importance of inclusive, community-led prevention efforts, similar to those described by Williams and colleagues [20]. Stakeholders focused on the goal of measuring those at risk for OUD (stage 0) because of the growing efforts to incorporate primary and secondary prevention interventions for this population within the treatment-oriented portions of the cascade of care. In addition, stakeholders acknowledged that movement through the cascade may not be linear, but will likely be cyclical given that both relapse and recovery occur in the context of OUD [36]. Having the ability to provide a specific target for the size of the population to be reached by system-level prevention efforts may lead to more effective means of engaging this population with both prevention and treatment services. The estimated size of this population (5.2% of people in Rhode Island) is similar to that reported in a recent study in the neighboring state of Massachusetts by Barocas and colleagues (4.6% in 2015) [51].

Initiation and continuous engagement with MOUD are critical steps in the cascade of care. Rhode Island has steadily increased capacity for MOUD enrollment as part of its Overdose Prevention and Intervention Action Plan, but full engagement with treatment programs continues to represent a statewide challenge. Prior work has identified system-level interventions to address this gap in care, including increasing low-barrier access and initiation of MOUD in

settings such as the emergency department [52,53]. Other interventions focus on increasing clinical training and education on buprenorphine prescription to increase the number of settings where people can initiate MOUD [52,54]. Based on preliminary estimates, it is estimated that about one-half of individuals with a diagnosis of OUD (stage 1) initiate MOUD (stage 2). This estimate is slightly higher than that reported by Larochelle and colleagues in the neighboring state of Massachusetts, where about one-third of individuals who experienced a nonfatal overdose later initiated methadone, buprenorphine, or naltrexone use in the following 12 months [55].

Stakeholders further identified the critical step of including and therefore measuring recovery-oriented systems of care and engagement with community-based recovery supports. There is additional value added to the model put forth by Williams and colleagues [15,20] by incorporating this stage as it captures a positive endpoint of OUD for many people. However, measuring recovery remains a challenge. The estimate we included in the initial parameterization for the cascade of care comes with some uncertainty. Engagement with recovery support services could be measured more precisely through referrals to community-based recovery supports such as peer recovery coaches, recovery-friendly employment programs, or recovery housing [37,56–58]. Measuring recovery or engagement with recovery support services is important for identifying interventions and capturing the protective factors associated with achieving and maintaining such a status, including the empowerment and resource capital that is inherent in recovery [38,59–63].

Currently, there is no existing population-based data source that allows us to measure the number of people who identify as being in recovery at the state level. The Behavioral Risk Factor Surveillance System, a national survey conducted annually by the Centers for Disease Control and Prevention, could add an optional module to achieve this aim [64]. The State of Oregon is using this approach by adding a module with items from the National Recovery Survey [37,56]. This approach offers a more consistent data source collected on an annual basis using the same validated items from the National Recovery Survey. In addition, states may consider implementing assessment of recovery at the clinical level by integrating annual screening tools into existing systems of care such as the Brief Assessment of Recovery Capital (BARC-10) [63,65,66], This scale offers a snapshot of 10 recovery measures that are indicative of someone's progression in their recovery from OUD [63,65,66]. Use of such a scale provides an opportunity to more accurately estimate the size of the population in the final stage of the cascade.

There are implications for how interventions may be applied to improve transitions between the stages, particularly if the goal is to maximize reduction in drug overdose deaths. Examples may include enhanced screening for OUD if a substantial drop between stages 0 and 1 is observed, increased initiation of MOUD for people diagnosed with OUD to address gaps between stages 1 and 2, deployment of case management interventions to improve retention in care to improve progress from stage 2 to stage 3, and providing funding for community-based recovery support services (e.g., housing, employment training programs) to move individuals from continuous use of MOUD (stage 3) to long-term recovery (stage 4) [52].

## Limitations

The primary goal of the current study was to develop and operationalize a statewide cascade of care for OUD, resulting in a "breadth of scope" hybrid model of unlinked data sources [26] offering a cross-sectional snapshot of the number of individuals in each stage. This limits our ability to fully understand the trajectories of specific individuals across all stages or account for true transitions from one stage to the next, particularly from stage 0 to stage 1 and from stage 3

to stage 4. For stage 0, we also recognize that there are limitations inherent in using self-reported measures of heroin use and prescription opioid misuse, including social desirability bias, where individuals who use heroin or misuse prescription opioids may report not doing so to avoid perceived judgment. Furthermore, recovery (stage 4) is often a self-identified status that is difficult to measure through administrative claims data due to potential overlap with other stages (i.e., those with long-term engagement with MOUD in stage 3) [67]. Future analyses will focus on improved linkages across datasets that account for movement between the outlying stages (stages 0 and 4) and the system of care transition stages (stages 1, 2, and 3), and the time-varying nature of the transition of individual patients from one stage to the next [67]. Given the challenges with unrelated, cross-sectional data sources, we ask that the descriptive data and results be interpreted with caution.

Similarly, the Rhode Island Cascade of Care for Opioid Use Disorder is not representative of detailed stages of OUD treatment nor is it inclusive of all available evidence-based treatment modalities. Our focus was to first establish the stages that stakeholders agreed upon, and for which we had access to the most complete data. For example, in stage 2 we used the number of unique initiations of MOUD statewide; however, this does not include individuals who were on MOUD while incarcerated [68]. Stages 2 and 3 also excluded extended-release injectable naltrexone, for which there were fewer than 100 records in Rhode Island in 2016. In addition, there are limitations with the current definition of recovery in stage 4 ("Did you used to have a problem with opioids and no longer do?") [38], as this may include individuals who would not meet the diagnostic criterion for OUD or may not have progressed through the cascade to reach this stage.

## Conclusion

The Rhode Island Cascade of Care for Opioid Use Disorder provides statewide data-sharing partners and policymakers with a starting point for understanding and assessing engagement in care, a system-level "snapshot" for prioritizing the datasets that would be useful in measuring the annual state of the system [14,17,26,56]. The 2 processes of undertaking a community-driven stakeholder process and prioritizing available datasets to populate this new model helped to operationalize and apply the cascade for the state. This process efficiently prioritized and identified the scope of the population and defined the essential stages of engagement with medication-based treatment for OUD. We succeeded in the original goal of operationalizing a population-level cascade orientated around promoting recovery, and we can now use this snapshot as a health policy tool to drive policy decisions and improve health outcomes across the state.

## Supporting information

**S1 STROBE Checklist.**
(DOC)

## Acknowledgments

We would like to thank the state agencies and departments involved in this work, in particular the Rhode Island Department of Health and the Rhode Island Executive Office of Health and Human Services, as well as the State of Rhode Island's Office of the Governor, for their feedback and support operationalizing the OUD cascade of care. We thank the Department of Behavioral Healthcare, Developmental Disabilities and Hospitals for its many contributions and for providing data from BHOLD. Additionally, we appreciate the support and feedback

received from members of the Governor's Overdose Prevention and Intervention Task Force and members of the Opioid Data Council working group in Rhode Island. Special thanks to those who contributed to this project, including Jamie Goulet; Linda Mahoney; Ryan Erickson; Catherine Gehring; Gabriella Arredondo, MPA; Meghan McCormick, MPH; Nicole Alexander Scott, MD, MPH; Philip Chan, MD, MS; and Laura C. Chambers, PhD, MPH. The opinions presented herein do not represent the views of the Centers for Disease Control and Prevention, nor do they represent the official policy of the Rhode Island Department of Health or other Rhode Island state agencies.

## Author Contributions

**Conceptualization:** Jesse L. Yedinak, Kimberly Paull, Rebecca Lebeau, Ashley L. Buchanan, Tom Coderre, Rebecca Boss, Josiah D. Rich, Brandon D. L. Marshall.

**Data curation:** Kimberly Paull, Rebecca Lebeau, Maxwell S. Krieger.

**Formal analysis:** William C. Goedel, Kimberly Paull, Rebecca Lebeau, Cheyenne Thompson, Brandon D. L. Marshall.

**Funding acquisition:** Brandon D. L. Marshall.

**Investigation:** Brandon D. L. Marshall.

**Methodology:** Jesse L. Yedinak, Ashley L. Buchanan, Tom Coderre, Rebecca Boss, Brandon D. L. Marshall.

**Project administration:** Jesse L. Yedinak.

**Supervision:** Josiah D. Rich.

**Visualization:** Maxwell S. Krieger.

**Writing – original draft:** Jesse L. Yedinak.

**Writing – review & editing:** Jesse L. Yedinak, William C. Goedel, Maxwell S. Krieger, Cheyenne Thompson, Ashley L. Buchanan, Tom Coderre, Rebecca Boss, Josiah D. Rich, Brandon D. L. Marshall.

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
