## [Decision Letter · Decision Letter 0]

25 Jul 2019

Dear Dr. Marshall,

Thank you very much for submitting your manuscript "Operationalizing a Statewide Care Continuum for Opioid Use Disorder: Identifying Gaps to Improve Care" (PMEDICINE-D-19-02173) for consideration at PLOS Medicine for our upcoming special issue on substance mis/use. 

Your paper was discussed among the editorial team, evaluated by the guest editors for the special issue, and sent to independent reviewers, including a statistical reviewer. The reviews are appended at the bottom of this email and any accompanying reviewer attachments can be seen via the link below:

[LINK]

In light of these reviews, we will not be able to accept the manuscript for publication in the journal in its current form, but we would like to invite you to submit a revised version that fully addresses the reviewers' and editors' comments. You will appreciate that we cannot make a decision about publication until we have seen the revised manuscript and your response, and we expect to seek re-review by one or more of the reviewers. 

We hope to receive your revised manuscript within two weeks. Please email us (plosmedicine@plos.org) if you have any questions or concerns.

Please let me know if you have any questions. Otherwise, we will look forward to receiving your revised manuscript shortly. 

Sincerely,

Richard Turner PhD, for Philippa Berman, MBBS

rturner@plos.org

In your data statement, are you able to name the "third party state institution"? If so, we would suggest including a contact point or person at this institution for readers who may wish to inquire about obtaining access to the relevant data. 

Please restructure your title so that the wording falling after the colon provides a study descriptor, e.g. "a cross-sectional study". 

Please combine the "methods" and "findings" subsections of your abstract. The final sentence of the new combined subsection should summarize the study's main limitations. 

After your abstract, we will need to ask you to add a new and accessible "author summary" section in non-identical prose. You may find it helpful to consult one or two recent research papers published in PLOS Medicine to get a sense of the preferred style. 

Please separate the "Methods" and "Results" sections of your main text. 

Early in your methods section, please state whether or not your study had a protocol or prespecified analysis plan, and if so attach the relevant document as a supplementary file. Please highlight all analyses that were not prespecified. 

Please add a completed checklist for the most appropriate reporting guideline (which we suspect may be STROBE or RECORD) as a supplementary document, and refer to this early in your methods section. In the checklist, please refer to individual elements by section (e.g. "methods") and paragraph number rather than by line or page numbers, as the latter generally change in the event of publication. 

To your methods section, please add a statement about ethics approval, if only to note that approval was judged unnecessary given the data sources. 

Are you able to add a table giving summary demographic (and relevant clinical and treatment) details for participants, early in your results section? If so, please also add a sentence on said demographic details to your abstract.

Please adapt all reference call-outs in your main text to the following style: "... treatment interventions [1].". Where multiple citations are made, please ensure that the square brackets contain no spaces. 

Please remove all instances of "[Internet]" from your reference list. 

Comments from the reviewers:

*** Reviewer #1: 

In general, I confine my remarks to statistical aspects of articles. There really weren't statistics here, but there was methodology. Unfortunately, I think it has some serious flaws.

First, it can't be both stages and a continuum. A continuum implies continuity. Stages deny that continuity.

Second, the authors did not show any evidence that people move from stage to stage, much less that they do so sequentially. Certainly people could go from stage 0 to stage 4 without the intermediate stages.

Third, stages, to me, implies exclusivity - but here, everyone is at stage 0 and some people are also at stage 1, 2, 3 or 4.

Fourth, the authors haven't really shown evidence that these stages exist - they certainly *might* exist and they make some sense, but why these particular definitions? Are these better than others? What other definitions were considered? How were these stages arrived at?

Fifth, many other people are at risk and risk isn't a yes/no variable. Different people are at different degrees of risk.

p. 7 It's not clear that anything in here is achievable. I certainly hope it is, but no evidence is presented.

p 7 You do not want people moving from stage 0 to stage 1.

p. 7 there is a typo of stage 5.

p. 9 Focusing solely on MAT isn't a good idea. Saying that you are doing so because the data are available is a bit like the famous drunk who looked for his keys under the lamppost rather than where he dropped them, because the light was better there. 

p. 11 For stage 0, what about people who deny use of opioids? Maybe stage 0 should be "denial".

p. 17 It is not really acceptable that stage 2 and 3 are defined so differently.

*** Reviewer #2: 

Review of PMEDICINE-D-19-02173, "Operationalizing a statewide care continuum for opioid use disorder: identifying gaps to improve care."

This manuscript modifies an existing model of a cascade of care for OUD and then populates the cascade cross-sectionally using a range of national and state-wide data sets for the state of Rhode Island. It is difficult for me to evaluate the manuscript's importance within the existing evidence on OUD risk, treatment uptake and retention, and recovery in the U.S., as I am not an expert in this area. I do, however, have substantial experience with cascades of care, for both HIV and NCDs. I will focus my comments in this area.

1. First, unlike the cascade of care of for HIV, to which the paper refers frequently, the stages defined for this cascade do not each represent a closed cohort. This is due largely to the definition of Stage 0 and how it is populated. As far as I can tell, there is no relationship between the data set used for Stage 0 and the data set used for Stage 1. There is therefore no reason that the population achieving Stage 1 (diagnosis) should overlap to any particular extent with the population defined in Stage 0 (risk). These could be entirely different people. It's likely that there is in fact a fair bit of overlap, but it's certainly possible that Stage 1 combines some people identified in Stage 0 and some newly identified at Stage 1. More important, as the manuscript states, many if not most people in Stage 0 will never be eligible for Stage 1. Dropoff between Stages 0 and 1 is thus not meaningful. That is not how a traditional cascade of care works—you normally start with a fixed denominator in the first stage and report proportions achieving each stage after that, with the goal of 100% moving from stage to stage. In this cascade, Stage 0 is problematic, and it tells us little or nothing about retention in care (since most people in it do not start care, and many not in it do start care). 

While measuring risk in the population is certainly important, it does not belong in this cascade of care, given the nature of OUD and the data available. I recommend removing Stage 0 and starting with Stage 1. A separate presentation of the proportion of those at risk whom you estimate reach Stage 1 would be fine.

2. There is no explanation of what we should expect from an OUD cascade of care. What proportions of patients are expected to reach each stage, based on what we know about treatment effectiveness and patient behavior? For non-experts, this information would place the results in context and allow interpretation.

3. While the concept of measuring program effectiveness and identifying gaps using a cascade is widely accepted and can be a valuable tool, populating it with unrelated cross-sectional data sets is problematic. Some of the data sets, such as claims data for medication, are probably quite accurate. Others, such as self-reported risk factors and self-reported recovery, are probably not. Moreover, some of the data sets represent national samples and others are state level. The authors mention these limitations, but they make no effort to consider the potential effect. At the very least, sensitivity analysis to consider the impact of higher and lower rates of achievement of each stage is needed. Ideally we'll one day have a prospective cohort that can follow individuals through the cascade; for now, it should be strongly emphasized that these are descriptive data and results and should be interpreted very cautiously.

4. I congratulate the authors on reporting proportions achieving each stage using the original denominator (Stage 0, though I would use Stage 1, as noted above). This is not done in HIV very often and can result in wildly misleading interpretations (e.g., reporting 95% virally suppressed, but only of those who know their status and started and treatment and had a viral load test—which is almost a guarantee of suppression—when in fact only half the infected population knows its status). That said, I think that this manuscript would benefit from including both—the proportion based on the original denominator and the proportion of those reaching the previous stage. The utility of the latter is that it does identify where we are losing people in the cascade. Figure 2 does not emphasize that the proportion lost between Stage 1 and Stage 2 is the most important point of dropoff and should thus likely be the main focus of intervention.

5. The explanation of where in the cascade people go who start treatment but drop off before 7 days is unclear.

6. The description of the role of the community stakeholders is also a little unclear. I get that it was due to the community that a last stage of recovery was defined as it was, but there is little explanation in the methods of who participated or exactly what they contributed.

7. Minor, but still important: the writing in this manuscript is sloppy. Many sentences are redundant, and there must be a hundred excess commas. There is no clear "Results" section. A good copy editor is needed.

8. The title of the article is misleading. The focus is not on identifying gaps, but on defining the cascade and populating it.

9. To end on a high note: the last sentence in the first paragraph headed "Defining each stage of the RI OUD care continuum" on page 8, which lists 3 purposes of the continuum, is really nice. If the paper could be structured around these three goals, it would be stronger.

*** Reviewer #3: 

This manuscript reflects an ambitious and important effort to quantify at the state level the engagement of people with opioid use disorder in care. The manuscript is clearly written and the methods are well described. The results have clear implications for improving the quality of services in Rhode Island and can be a model for other states to evaluate their own services for people with opioid use disorder.

It is a strength of this analysis that the authors include a Stage 4 for recovery and define recovery broadly, but there are inconsistencies in the way Stage 4 is presented in the paper that are confusing. In the Methods Section on p10, Stage 4 is defined in several ways, including both self-reported recovery (through any possible supportive pathway) and also sustained retention (i.e. >180 days) in medication treatment. This is confusing because Stage 3 is also defined by >180d in treatment without significant interruption. In the analysis, recovery is only defined by self report from a national survey. It seems that there are opportunities to use administrative data, including those with sustained uninterrupted retention in treatment, to include people on medications in Stage 4, as the authors suggest is possible in the Methods Section. Similarly, in the discussion of Stage 4 on p18, the authors omit consideration of ways to include sustained retention in treatment as a part of Stage 4.

There is no consideration of how statistical uncertainty should be incorporated into the estimates presented. While quantifying uncertainty might not be indicated for estimates from a complete sample of the population (e.g. the complete PDMP database), uncertainty estimates should be included around point estimates from probability samples (specifically, the National Survey on Drug Use and Health and the National Recovery Survey). 

A major limitation of the analysis is the exclusion of people retained in buprenorphine treatment from Stage 3, though this limitation is highlighted and discussed in the manuscript. The authors discuss how gaps between prescriptions exclude many people on buprenorphine from the meeting criteria for Stage 3 and they present the mean number of days between prescriptions and average number of refills per year. It would be more instructive to also present the number of people on buprenorphine who met criteria for Stage 3 (no treatment interruptions >7 days or some equivalent formulation based on number of fills) and number that did not. The authors could then justify whether or not to include these people in the total for Stage 3. 

A minor point from the discussion: When stating that RI has high rates of initiation and retention in medication treatment for OUD, it would be helpful to reference what your standard is for determining "high" (e.g. rates from other states).

- 

Alexander R. Bazazi, MD PhD

***

[LINK]

---

## [Decision Letter · Decision Letter 1]

27 Sep 2019

Dear Dr. Marshall,

Thank you very much for re-submitting your manuscript "Defining a Recovery-Oriented Cascade of Care for Opioid Use Disorder: A Community-Driven, Statewide Cross-Sectional Assessment" (PMEDICINE-D-19-02173R1) for consideration at PLOS Medicine for our upcoming special issue on substance mis/use.

I have discussed the paper with editorial colleagues and our academic editor, and it was also seen again by three reviewers. I am pleased to tell you that, provided the remaining editorial and production issues are dealt with, we expect to be able to accept the paper for publication in the journal.

[LINK]

Please let me know if you have any questions. Otherwise, we look forward to receiving your revised manuscript shortly. 

Kind regards,

Richard Turner PhD, for Philippa Berman, MBBS

rturner@plos.org

Requests from Editors:

Around line 41, please cite the date that the study was done, and add a few words to summarize the groups of stakeholders involved. 

At line 51, please quote at least one further limitation, e.g. use of self-report in arriving at estimates. 

At line 55, please add "Our findings indicate that cross-sectional summaries ...", or similar, at the start of the sentence. 

Around line 160, please add a few words to state explicitly that the study did not have a prespecified plan. 

Around lines 170-75, please convert "percent of" to "proportion of". 

Please refer to the attached STROBE checklist in the methods section of your main text. 

Please ensure that all references match journal format. In reference 50, for example, the journal name needs to be abbreviated consistently with other citations, and the volume number adding. 

In reference 13, please remove the superfluous "g". 

Please spell out "Nqf" in reference 43. 

We suggest substituting a panel for figure 1, which should suffice to define the individual stages. 

As suggested by one referee, please adapt figure 2 to add to or adapt the arrows to indicate additional possible participant pathways. 

Please remove the logos from figure 2. 

Please adapt your STROBE checklist so that individual items are referred to by section (e.g. "Methods") and paragraph number rather than by page or line numbers, as the latter generally change upon publication. 

Comments from Reviewers:

*** Reviewer #1: 

I am checking "proceed without recommendation" because I do not know what to do here.

The authors did respond to each of my comments - thank you! - but I am not at all convinced the end result does what it claims to do. However, since the reasons for my concerns are more substantive than statistical, I am going to leave it to other reviewers and the editors to decide what to do from here. I'm not expert enough to render a final judgment on this.

Peter Flom

*** Reviewer #2: 

Overall this is a much stronger manuscript than the original version. The additions and revisions made by the authors do a good job of addressing most concerns, and I congratulate them for it. In particular, the description of the stakeholder engagement process is now excellent, and I am glad to see reported the incremental proportions achieving each stage (personally I'd add these to the figure as well, but reporting them in the text is sufficient).

I still have strong reservations about the inclusion of Stage 0, though the authors have greatly improved how this stage is presented. My own preference remains that it be disconnected from the stage and, if desired, described separately, so that the cascade starts at current stage 1. I will defer to the editors' preferences on this, however. If Stage 0 is retained, I'd encourage a revision of Figure 1 to show that people can drop out of stage 0 entirely, and perhaps that they can move directly from stage 1 or stage 2 to stage 4. Cascades are often illustrated with arrows to show non-linear movement or losses to follow up; in this case, such a presentation would be helpful.

Other comments:

Author summary: Statement "Using national survey estimates and statewide administrative claims databases, we found that 47,000 Rhode Islanders were at risk for OUD (Stage 0) in 2016 and about half were diagnosed (Stage 1)" is misleading, as it implies that everyone in Stage 0 should be diagnosed but only half are. Please just state how many are in Stage 1, without the implication that everyone moves from Stage 0 to Stage 1.

Author summary: I'm also uneasy about the statement, "Engagement with a diverse group of stakeholders can result in the development of a cascade of care to assess and measure the success of statewide health systems in delivering interventions that reduce the number of individuals at risk for OUD and increase the numbers of individuals with OUD who are able to achieve long-term recovery." It's a complicated sentence that could easily be understood to mean that developing a cascade of care is what reduces number at risk and increases numbers who recover. I'd rephrase it if possible.

Lines 233-236: I do not understand the statement, "The cascade includes a primary prevention goal of reducing the number of individuals at risk for OUD (Stage 0), while also increasing the number who remain active in their recovery through engagement with professional and peer-based support services (Stage 4)." The cascade is a tool for presenting information. The goals mentioned are goals of the services and service providers, not the cascade.

There are still some writing errors, e.g. in line 760 "criteria" should be criterion, and in line 766 the newly added "to" should be dropped. PLOS Med's copy editor may catch these.

*** Reviewer #3: 

The authors have adequately addressed my concerns in their revisions.

***

[LINK]

---

## [Editor Report · Decision Letter 2]

14 Oct 2019

Dear Dr. Marshall, 

On behalf of my colleagues and the academic editor, Dr. Margarita Alegria, I am delighted to inform you that your manuscript entitled "Defining a Recovery-Oriented Cascade of Care for Opioid Use Disorder: A Community-Driven, Statewide Cross-Sectional Assessment" (PMEDICINE-D-19-02173R2) has been accepted for publication in PLOS Medicine. 

PRODUCTION PROCESS

PRESS

PROFILE INFORMATION

Thank you again for submitting the manuscript to PLOS Medicine. We look forward to publishing it. 

Best wishes, 

Richard Turner

Senior Editor 

PLOS Medicine

plosmedicine.org